# Vibration Analysis and Active Control of Rotor Shaft in Magnetically Suspended Air-Blower

**Lingbo Zheng [1], Wansheng Nie [1] and Biao Xiang [2,\*]**

[1] Department of Aerospace Science and Technology, Space Engineering University, Beijing 101416, China; zlb165@163.com (L.Z.); nws1969@126.com (W.N.)

[2] School of Mechano-Electronic Engineering, Xidian University, Xi'an 710071, China

\* Correspondence: xiangbiao@xidian.edu.cn

**Abstract:** An air-blower with active magnetic bearings could improve working efficiency and reduce energy consumption by avoiding contact between the rotor shaft and the stator part. The structure and prototype of a magnetically suspended air-blower are herein introduced, and the force models of active magnetic bearings developed. Furthermore, the dynamic models of a rotor shaft with unbalance terms were established to investigate the vibration characteristics of the magnetically suspended air-blower. The vibration characteristics of the rotor shaft with unbalance terms were analyzed, and the complex-field cross-feedback control was designed to suppress the vibration amplitude. Finally, experiments were conducted to verify the theoretical models, and the results indicated that the vibration amplitude of the rotor shaft with unbalance terms could be intensified by the rotating frequency, and the nutation vibration was reduced by 50% through increasing the high-frequency nutation coefficient of the complex-field cross-feedback control model. The results indicated that the vibration analysis of the rotor shaft was meaningful to the design and control of the magnetically suspended air-blower.

**Keywords:** magnetically suspended air-blower; unbalance term; complex-field cross-feedback control; nutation vibration; vibration characteristic

## 1. Introduction

Compared to normal rotary machines supported by mechanical bearing, the magnetically suspended rotary machine with stable suspension of active magnetic bearing (AMB) [1,2] has advantages of contactless suspension, free-lubrication, micro-vibration and active controllability. Therefore, to improve the dynamic performance and service period of rotary machines, the AMB has been widely used in many kinds of rotary machines, such as the momentum flywheel for satellite altitude control [3,4], the flywheel energy storage system (FESS) [5,6] and the high energy density motor [7,8].

However, for magnetically suspended rotary machines, the unbalance terms of the rotor shaft still lead to vibration behavior of the magnetically suspended rotary machine, which then disturbs the suspension stability and position precision of the rotor shaft [9,10]. Therefore, vibration analysis and active control of magnetically suspended rotary machines are worthy of being researched. For the magnetically suspended motor of FESS, the vibration characteristics of a flywheel rotor with a great inertial moment were researched in [11]. The results indicated that the stiffness coefficient of AMB could change the natural frequency of the flywheel rotor, and the damping coefficient of AMB could suppress the vibration amplitude of the rotor shaft at different rotating frequencies. The FESS with self-bearing hysteresis drive was proposed for the satellite energy system, and the vibration performance of the satellite flywheel rotor was tested. The results showed that the magnetic suspension system could satisfy performance at the maximum speed of

the flywheel rotor [12]. However, the effective vibration control method of the rotor shaft was not designed for the FESS. In addition, the vibration analysis method of the magnetically suspended flywheel rotor for altitude control of satellites was researched in [13], and the balancing method was designed to mitigate the vibration amplitude of the flywheel rotor. The vibration performance of the magnetically suspended flywheel was measured, and then the test rig and subcomponents were used to govern the dynamics of the flywheel rotor. The results implied that the maximum displacement of the flywheel rotor was reduced by 37% [14], but the active control method for the nutation vibration and precession vibration were not studied. The vibration model of a magnetically suspended motor with slim rotor was studied in [15], and a novel notch filter was designed for suppressing the harmonic vibration of the slim rotor. The experimental results showed that the vibration magnitude of the slim rotor was reduced by at least one order. The vibration suppression method of active and passive hybrid bearing-less motor was researched, and the axial vibration of the rotor shaft could be suppressed by the displacement terms estimated by the flux linkage [16]. The vibration of the rotor shaft was suppressed by the magnetic actuator when the rotating speed exceeded 9000 rpm, and the results presented showed that the vibration amplitude was reduced over 70% compared to the mechanical rotor [17]. Nevertheless, the theoretical vibration model of the magnetically suspended motor was not developed. Moreover, some active control methods were also used to suppress the vibration of the magnetically suspended rotor. For example, the variable angle compensation model [18], adaptive control [19,20] and optimal control [21] were respectively designed to suppress the synchronous vibration of the magnetically suspended rotor, and the results proved that the proposed control models were useful to mitigate the synchronous vibration of the magnetically suspended rotor by measuring the displacement variation and the power spectrum density. However, the vibration amplitudes at other frequencies, like the nutation vibration and precession vibration, were not effectively suppressed, and the unbalance terms of the magnetically suspended rotor were not considered.

Therefore, the vibration characteristics of the rotor shaft in a magnetically suspended air-blower with unbalance terms were investigated in this article, and then the complex-field cross-feedback control model was designed to suppress the nutation vibration and the precession vibration of the rotor shaft when the rotating speed exceeded 10,000 rpm. The contributions of this article are listed as follows:

(1) The vibration models of the magnetically suspended air-blower with unbalance terms were developed, and the current model of AMB-rotor shaft with unbalance vibration was established.

(2) The complex-field cross-feedback control model was designed to suppress the nutation and precession vibration of magnetically suspended air-blower.

(3) The vibration analysis was critical to structure design and the active control high-speed rotor suspended by the magnetic forces, and it was fundamental to the vibration control of the air-blower using magnetic forces.

This article is organized as follows. The structure and prototype of the magnetically suspended air-blower are introduced in Section 2. The dynamic models of the rotor shaft with magnetic forces and unbalance terms are developed in Section 3. Furthermore, in Section 4, the vibration characteristics of the rotor shaft with unbalance terms are investigated. The experiments about the theoretical model are conducted in Section 5. Finally, conclusions are drawn on the vibration characteristics and active control of the rotor shaft in the magnetically suspended air-blower.

## 2. Structure of Magnetically Suspended Air-Blower

### 2.1. The Prototype of Magnetically Suspended Air-Blower

The prototype of the magnetically suspended air-blower is illustrated in Figure 1, the whole system includes axial thrust AMB, radial AMB at A-end and B-end of rotor shaft, auxiliary bearings at A-end and B-end, permanent magnet synchronous motor (PMSM) and displacement sensors at A-end and B-end. In detail, the axial thrust AMB generates magnetic forces to overcome gravity, and then the rotor shaft would be suspended at the axial equilibrium position to avoid mechanical friction. In addition, the radial AMB at A-end and B-end of rotor shaft could generate magnetic forces to make the rotor shaft suspend at the radial equilibrium position, and the torques between the magnetic force at A-end and at B-end would control the radial torsions of the rotor shaft. The auxiliary bearing at A-end and B-end could protect the rotor shaft from excessive displacement deviation when the radial and axial AMBs are invalid. As the drive unit of the magnetically suspended air-blower, the PMSM could turn the rotational speed of the rotor shaft. The eddy current displacement sensors at the A-end and B-end of rotor shaft could measure the displacement variations in radial directions, and the axial eddy current displacement sensors below the thrust disc could measure the displacement variation of the rotor shaft in axial direction.

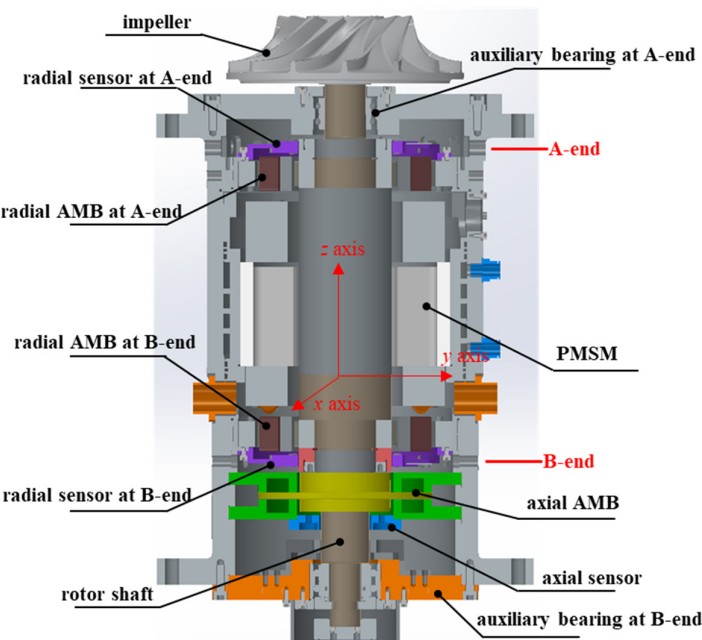

**Figure 1.** The prototype of magnetically suspended air-blower.

### 2.2. The Force Modeling of Radial AMB

As the core component of the magnetically suspended air-blower, the radial AMB in Figure 2 could realize the stable suspension of the rotor shaft in radial directions, and also actively regulate the suspension positions of the rotor shaft. As shown in Figure 2a, the magnet poles of the radial AMB consist of the bias magnet pole and the control bias magnet pole. The bias magnet pole with bias current could generate magnetic force to accomplish the suspension of the rotor shaft, and the control magnet pole with control current could regulate the suspension position of the rotor shaft when there are disturbances acting on it. As a consequence, the suspension position and displacement variation of the rotor shaft along $x$ axis and $y$ axis are actively controlled by the radial AMB.

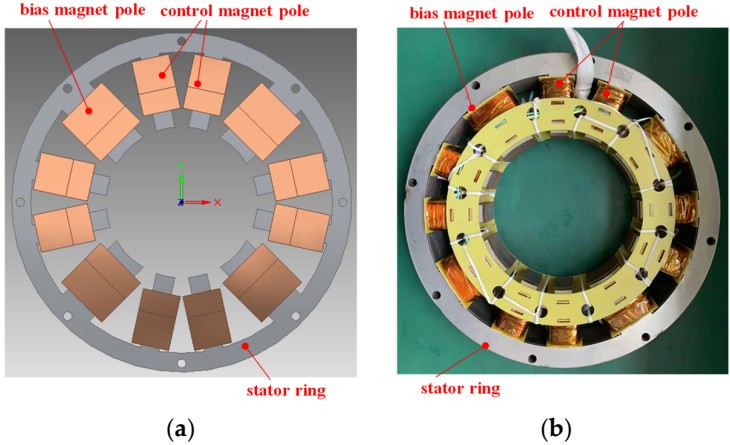

**Figure 2.** (**a**)The prototype of radial AMB, (**b**) the stator part of radial AMB.

According to the magnet flux distribution of the radial AMB in Figure 3a, the magnet flux generated by the electromagnetic (EM) windings passes the stator magnet pole, airgap, rotor magnet pole and the magnet ring, and finally returns back to the stator magnet pole. Furthermore, the equivalent magnetic circuits of the radial AMB are shown in Figure 3b, and magnet circuits could be separated into four decoupling loops along $x+$, $x-$, $y+$ and $y-$ directions. $R_c$ is the equivalent magnet reluctance of the control magnet pole and airgap, and $R_b$ is the equivalent magnet reluctance of the bias magnet pole and airgap. $N_c$ is the number of turns of the control magnet pole, and $N_b$ is the number of turns of the bias magnet pole. $I_c$ is the control current of the control magnet pole, $I_b$ is the control current of the bias magnet pole, and subscripts $\{x+, x-, y+, y-\}$ indicate the different directions and axes.

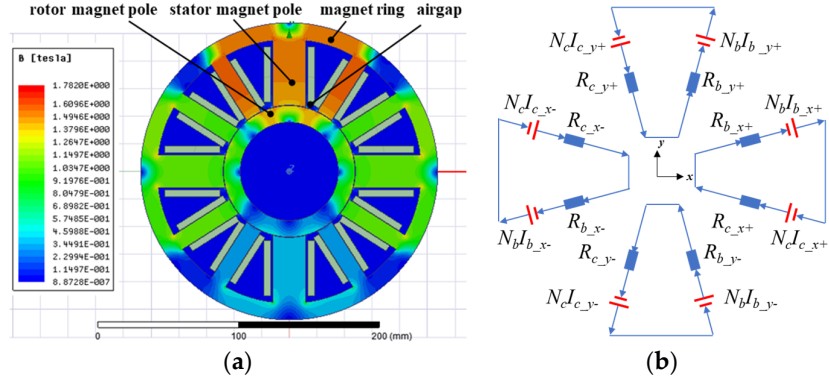

**Figure 3.** (**a**) The magnet flux distribution of radial AMB, (**b**) the equivalent magnetic circuits of radial AMB.

Therefore, based on the equivalent magnetic circuit model [22], the equivalent magnet reluctances of radial AMB are expressed into

$$
\begin{cases}
R_{y+} = R_{b\_y+} + R_{c\_y+} = \dfrac{2d_{y+}}{\mu_0 A_{y+}}; \quad R_{y-} = R_{b\_y-} + R_{c\_y-} = \dfrac{2d_{y-}}{\mu_0 A_{y-}} \\[2mm]
R_{x+} = R_{b\_x+} + R_{c\_x+} = \dfrac{2d_{x+}}{\mu_0 A_{x+}}; \quad R_{x-} = R_{b\_x-} + R_{c\_x-} = \dfrac{2d_{x-}}{\mu_0 A_{x-}}
\end{cases}
\tag{1}
$$

where $d$ with subscripts $\{x+, x-, y+, y-\}$ is the airgap length between the rotor magnet pole and the stator magnet pole along different axes of radial AMB, and $A$ with subscripts $\{x+,$

$x-$, $y+$, $y-$}is the cross-sectional area of magnet pole in different axes of radial AMB, $\mu_0$ is the vacuum permeability.

Considering the flux leakage is $\sigma$, the magnet flux along different directions of radial AMB are written into

$$\begin{cases} \Phi_{y+} = \dfrac{N_b I_{b\_y+} + N_c I_{c\_y+}}{\sigma R_{y+}}; \quad \Phi_{y-} = \dfrac{N_b I_{b\_y-} + N_c I_{c\_y-}}{\sigma R_{y-}} \\ \Phi_{x+} = \dfrac{N_b I_{b\_x+} + N_c I_{c\_x+}}{\sigma R_{x+}}; \quad \Phi_{x-} = \dfrac{N_b I_{b\_x-} + N_c I_{c\_x-}}{\sigma R_{x-}} \end{cases} \tag{2}$$

Therefore, the magnetic forces of radial AMB in different directions are obtained as follows

$$\begin{cases} f_{y+} = \dfrac{\Phi_{y+}^2}{\mu_0 A_{y+}}; \quad f_{y-} = \dfrac{\Phi_{y-}^2}{\mu_0 A_{y-}} \\ f_{x+} = \dfrac{\Phi_{x+}^2}{\mu_0 A_{x+}}; \quad f_{x-} = \dfrac{\Phi_{x-}^2}{\mu_0 A_{x-}} \end{cases} \tag{3}$$

Based on the equivalent magnetic circuits of radial AMB in Figure 3b, the magnet flux of radial AMB consists of control magnet flux and bias magnet flux, and the differential control model is used to the radial AMB in $x$ axis and $y$ axis. Therefore, the bias current of EM winding in the same direction is defined as the same value ($I_{b\_y+} = I_{b\_y-}$ and $I_{b\_x+} = -I_{b\_x-}$), and the control currents of EM winding are set as $I_{c\_y+} = -I_{c\_y-}$ and $I_{c\_x+} = -I_{c\_x-}$. Moreover, the area of magnet pole in radial direction is $A_{x+} = A_{x-} = A_x$ and $A_{y+} = A_{y-} = A_y$.

The displacement terms between the stator magnet pole and the rotor magnet pole in different directions are

$$\begin{cases} d_{y+} = d_{y0} + d_y; \quad d_{y-} = d_{y0} - d_y \\ d_{x+} = d_{x0} + d_x; \quad d_{x-} = d_{x0} - d_x \end{cases} \tag{4}$$

where $d_{y0}$ and $d_{x0}$ is the bias airgap in $y$ axis and $x$ axis, $d_y$ and $d_x$ is the control airgap in $y$ axis and $x$ axis.

Based on the Biot-Savart law [23], the resultant magnetic forces of radial AMB along $x$ axis and $y$ axis are

$$\begin{cases} f_y(I_{c\_y}, d_y) = \dfrac{\mu_0 A_y}{4\sigma^2} \left[ \dfrac{(N_b I_{b\_y} + N_c I_{c\_y})^2}{(d_{y0} - d_y)^2} - \dfrac{(N_b I_{b\_y} - N_c I_{c\_y})^2}{(d_{y0} + d_y)^2} \right] \\ f_x(I_{c\_x}, d_x) = \dfrac{\mu_0 A_x}{4\sigma^2} \left[ \dfrac{(N_b I_{b\_x} + N_c I_{c\_x})^2}{(d_{x0} - d_x)^2} - \dfrac{(N_b I_{b\_x} - N_c I_{c\_x})^2}{(d_{x0} + d_x)^2} \right] \end{cases} \tag{5}$$

given the turn of number $N_c = N_b$, (5) could be expressed into

$$\begin{cases} f_y(I_{c\_y}, d_y) = \dfrac{\mu_0 A_y N_b^2}{4\sigma^2} \left[ \dfrac{(I_{b\_y} + I_{c\_y})^2}{(d_{y0} - d_y)^2} - \dfrac{(I_{b\_y} - I_{c\_y})^2}{(d_{y0} + d_y)^2} \right] \\ f_x(I_{c\_x}, d_x) = \dfrac{\mu_0 A_x N_b^2}{4\sigma^2} \left[ \dfrac{(I_{b\_x} + I_{c\_x})^2}{(d_{x0} - d_x)^2} - \dfrac{(I_{b\_x} - I_{c\_x})^2}{(d_{x0} + d_x)^2} \right] \end{cases} \tag{6}$$

with the derivative function of magnetic force in (6), the stiffness coefficients of radial AMB are

$$\begin{cases} k_{iy} = \dfrac{\partial f_y(I_{c\_y}, d_y)}{\partial I_{c\_y}} = \dfrac{\mu_0 A_y N_c}{\sigma^2} \dfrac{N_c I_{c\_y}\left(d_{y0}^2 + d_y^2\right) + N_b I_{b\_y} d_{y0} d_y}{\left(d_{y0} + d_y\right)^2 \left(d_{y0} - d_y\right)^2} \\[4mm] k_{dy} = \dfrac{\partial f_y(I_{c\_y}, d_y)}{\partial d_y} = \dfrac{\mu_0 A_y N_c}{2\sigma^2} \dfrac{(N_b I_{b\_x} + N_c I_{c\_x})^2 \left(d_{y0} + d_y\right)^3 + (N_b I_{b\_x} - N_c I_{c\_x})^2 \left(d_{y0} - d_y\right)^3}{\left(d_{y0} + d_y\right)^3 \left(d_{y0} - d_y\right)^3} \end{cases} \quad (7)$$

When the rotor shaft of the magnetically suspended air-blower is located on the protective bearing in radial direction, the bias displacement in the radial direction is −0.2 mm, and then the magnetic forces of radial AMB with −0.2 mm bias displacement and 1.5 A bias current are plotted by the red line in Figure 4. The magnetic force is increased with the control current, and the maximum force is about 160 N at the 1.5 A control current. Moreover, when the rotor shaft of the magnetically suspended air-blower is at the equilibrium position in radial direction, the bias displacement is 0 mm, and then the magnetic forces of radial AMB with 0 mm bias displacement and 1.5 A bias current are shown by the blue line in Figure 4. The maximum value of magnetic force at 1.5 A control current is 910 N. Therefore, the magnetic force of radial AMB is increased with the control current, and decreased with the displacement.

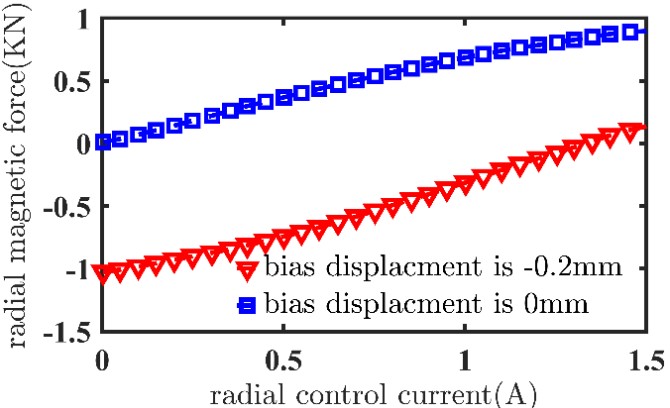

**Figure 4.** The magnetic forces of radial AMB with the bias current 1.5 A.

### 2.3. The Force Modeling of Axial AMB

To overcome gravity for the rotor shaft, the axial thrust AMB in Figure 5 was designed to generate the magnetic force along the axial direction. As shown in Figure 5a, the axial AMB consisted of upper EM winding, lower EM winding and a thrust disc mounted on the rotor shaft, and the EM winding is illustrated in Figure 5b. Moreover, based on the displacement measurement of the axial sensor, the differential control model of upper EM winding and lower EM winding were designed to control the axial displacement of the rotor shaft. The control current of the upper and lower EM winding would be reduced when the displacement of the rotor shaft deflected to the positive value along the axial axis, and the control current would be increased when the displacement of the rotor shaft varied to the negative value on the axial axis.

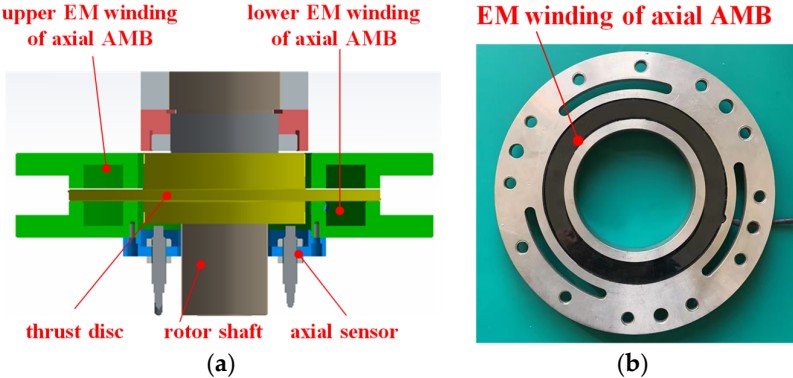

**Figure 5.** (**a**) The prototype of axial AMB, (**b**) the stator winding of axial AMB.

The magnetic flux paths of axial AMB are plotted in Figure 6, for the magnetic path of upper EM winding, the magnetic flux passes the upper stator magnet pole, upper airgap, the rotor magnet pole, and finally returns back to the EM winding. For the magnetic path of upper EM winding, the magnetic flux passes the lower stator magnet pole, lower airgap, rotor magnet pole, and goes back to EM winding. Furthermore, the equivalent magnetic circuits of axial AMB are plotted in Figure 6b. $R_e$ with subscripts {$z+$, $z-$} indicates the equivalent magnetic reluctance of external airgap, $R_i$ with subscripts {$z+$, $z-$} indicates the equivalent magnetic reluctance of interior airgap, $R_{z+}$ and $R_{z-}$ are the equivalent magnetic reluctances of stator and rotor magnet poles at upper and lower ends; respectively.

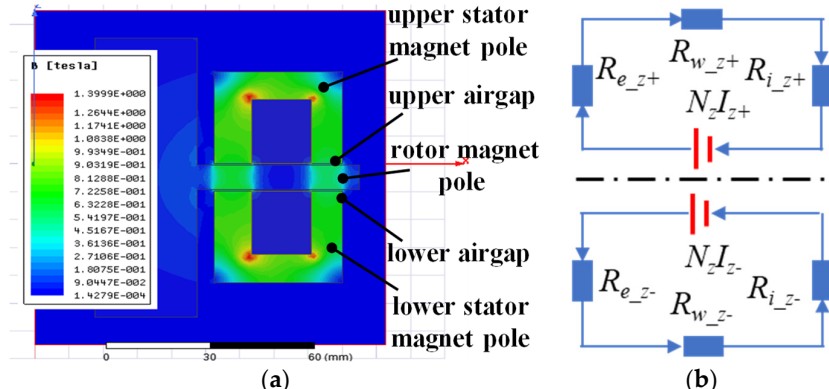

**Figure 6.** (**a**) The magnet flux distribution of axial AMB, (**b**) the equivalent magnetic circuits of axial AMB.

The equivalent magnetic reluctances could be expressed into

$$\begin{cases} R_{e\_z+} = \dfrac{d_{z+}}{\mu_0 A_{e\_z+}}; \; R_{i\_z+} = \dfrac{d_{z+}}{\mu_0 A_{i\_z+}}; \; R_{z+} = \dfrac{d_{z+}}{\mu_0 A_{z+}} \\ R_{e\_z-} = \dfrac{d_{z-}}{\mu_0 A_{e\_z-}}; \; R_{i\_z-} = \dfrac{d_{z-}}{\mu_0 A_{i\_z-}}; \; R_{z-} = \dfrac{d_{z-}}{\mu_0 A_{z-}} \end{cases} \tag{8}$$

where $d_{z+}$ and $d_{z-}$ are the upper and lower displacements between the stator magnet pole and the rotor magnet pole, and $A_e$ with subscripts {$z+$, $z-$} are the exterior area of magnet pole at upper and lower end, and $A_i$ with subscripts {$z+$, $z-$} are the interior area of magnet pole at upper and lower end, and $A_{z+}$ and $A_{z-}$ are the area of magnet pole at upper and lower end.

The magnet flux along positive and negative directions of axial AMB are

$$\begin{cases} \Phi_{z+} = \dfrac{N_z I_{z+}}{\sigma\left(R_{e\_z+} + R_{i\_z+} + R_{z+}\right)} \\[3mm] \Phi_{z-} = \dfrac{N_z I_{z-}}{\sigma\left(R_{e\_z-} + R_{i\_z-} + R_{z-}\right)} \end{cases} \tag{9}$$

Using the differential control model, the displacements and currents of axial AMB are written into

$$\begin{cases} d_{z+} = d_{z0} - d_z; \quad d_{z-} = d_{z0} + d_z \\ I_{z+} = I_{b\_z} + I_{c\_z}; \quad I_{z-} = I_{b\_z} - I_{c\_z} \end{cases} \tag{10}$$

where $d_{z0}$ is the bias displacement in axial direction, and $d_z$ is the control displacement, $I_{b\_z}$ is the bias current of axial EM winding, and $I_{c\_z}$ is the control current.

The magnetic forces generated by axial AMB at the exterior and interior airgaps of lower and upper end are

$$\begin{cases} f_{e\_z+} = \dfrac{\Phi_{z+}^2}{2\mu_0 A_{e\_z+}}; \; f_{i\_z+} = \dfrac{\Phi_{z+}^2}{2\mu_0 A_{i\_z+}} \\[3mm] f_{e\_z-} = \dfrac{\Phi_{z-}^2}{2\mu_0 A_{e\_z-}}; \; f_{i\_z-} = \dfrac{\Phi_{z-}^2}{2\mu_0 A_{i\_z-}} \end{cases} \tag{11}$$

For the axial AMB, there are $A_z = A_{e\_z+} = A_{i\_z+} = A_{e\_z-} = A_{i\_z-}$, and then the final magnetic force in axial direction is expressed into

$$f_z(I_{c\_z}, d_z) = f_{e\_z+} + f_{i\_z+} - f_{e\_z-} - f_{i\_z-} = \frac{\mu_0 A_z N_z^2}{9\sigma^2}\left[\frac{\left(I_{b\_z} + I_{c\_z}\right)^2}{\left(d_{z0} - d_z\right)^2} - \frac{\left(I_{b\_z} - I_{c\_z}\right)^2}{\left(d_{z0} + d_z\right)^2}\right] \tag{12}$$

The stiffness coefficients of axial AMB are written into

$$\begin{cases} \dfrac{\partial f_z}{\partial d_z} = \dfrac{2\mu_0 A_z N_z^2}{9\sigma^2}\dfrac{\left(I_{b\_z} + I_{c\_z}\right)^2\left(d_{z0} + d_z\right)^3 + \left(I_{b\_z} - I_{c\_z}\right)^2\left(d_{z0} - d_z\right)^3}{\left(d_{z0} - d_z\right)^3\left(d_{z0} + d_z\right)^3} \\[4mm] \dfrac{\partial f_z}{\partial i_z} = \dfrac{2\mu_0 A_z N_z^2}{9\sigma^2}\dfrac{\left(I_{b\_z} + I_{c\_z}\right)\left(d_{z0} + d_z\right)^2 - \left(I_{b\_z} - I_{c\_z}\right)\left(d_{z0} - d_z\right)^2}{\left(d_{z0} - d_z\right)^2\left(d_{z0} + d_z\right)^2} \end{cases} \tag{13}$$

The magnetic forces of the axial AMB with different bias displacements are plotted in Figure 7. When the rotor shaft is located on the protective bearing in the axial direction, the bias displacement of the rotor shaft in the axial direction is −0.2 mm, and the magnetic force of axial AMB is plotted by the red line. It increased with the control current, and the maximum value of magnetic force at 3 A control current is 2138 N. In addition, when the rotor shaft was suspended at the equilibrium position in the axial direction, the bias displacement in the axial direction is 0 mm, the magnetic force of axial AMB is shown by the blue line, and the maximum value at 3 A control current is 4687 N. When the bias displacement of the rotor shaft in the axial direction is 0.05 mm, the magnetic force is marked by the green line, and the maximum value is about 5553 N at 3 A control current. Therefore, the magnetic force of the axial AMB could be improved by regulating the bias displacement and the control current.

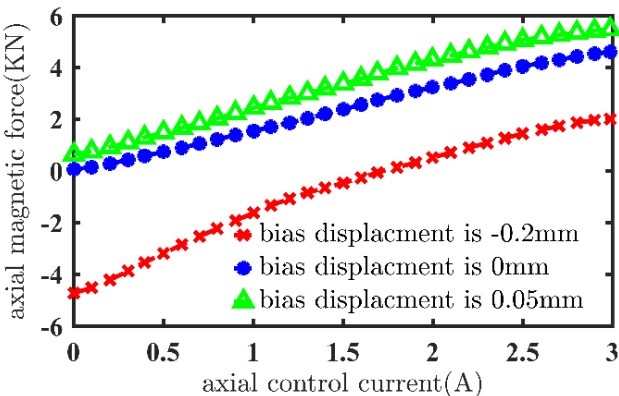

**Figure 7.** The magnetic forces of axial AMB with the bias current 1.5 A.

## 3. Modeling of Rotor Shaft in Magnetically Suspended Air-Blower

### 3.1. Force Model of Rotor Shaft with Unbalance Terms

The force models of the magnetically suspended air-blower on radial DOFs are shown in Figure 8. The magnetic forces ($f_{xa}$, $f_{ya}$, $f_{xb}$, $f_{yb}$) at the A-end and B-end control radial suspension and tilting of rotor shaft are shown. In Figure 8a, the distance from the mass of center to the radial AMB at A-end is $l_{aa}$, and the distance from the mass of center to the radial AMB at B-end is $l_{ab}$. Moreover, the generalized coordinate of the rotor shaft is $[x, \beta, y, -\alpha]^T$, and the coordinate of the radial AMB is $\mathbf{q}_m = [x_{ax}, x_{bx}, y_{ay}, y_{by}]^T$, and the coordinate of displacement sensor is $\mathbf{q}_s = [s_{ax}, s_{bx}, s_{ay}, s_{by}]^T$. In Figure 8b, $C_i$ is the center position of the inertial principal axis, and $C_g$ is the center position of the geometrical principal axis. Furthermore, the displacement of the inertial principal axis is $\mathbf{q}_i = [x_i, \beta_i, y_i, -\alpha_i]^T$, and the displacement of the geometrical principal axis is $\mathbf{q}_g = [x_g, \beta_g, y_g, -\alpha_g]^T$.

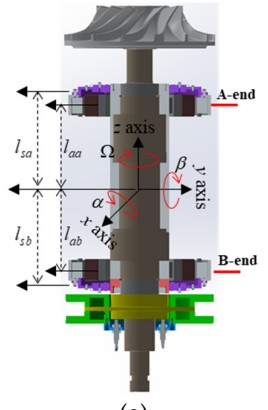 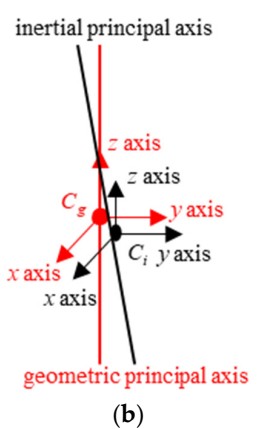 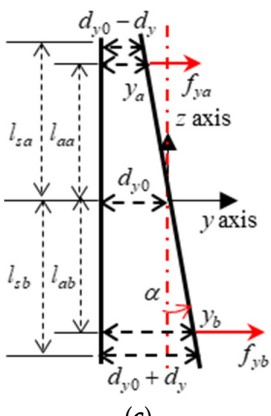 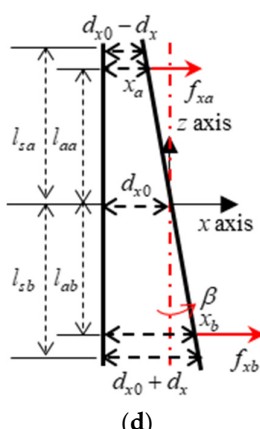

(**a**)      (**b**)      (**c**)      (**d**)

**Figure 8.** (**a**) The geometrical structure of magnetically suspended air-blower, (**b**) the inertial principal axis and geometric principal axis of magnetically suspended air-blower, (**c**) the rotating of rotor shaft around $x$ axis, (**d**) the rotating of rotor shaft around $y$ axis.

The relationship between the coordinates of radial AMB and displacement sensor are expressed into

$$\begin{cases} \mathbf{q}_m = \mathbf{T}_f^T \mathbf{q}_g \\ \mathbf{q}_s = \mathbf{T}_s \mathbf{q}_g \end{cases} \tag{14}$$

where $\mathbf{T}_f = \begin{bmatrix} 1 & 1 & 0 & 0 \\ l_{ab} & -l_{aa} & 0 & 0 \\ 0 & 0 & 1 & 1 \\ 0 & 0 & -l_{ab} & l_{aa} \end{bmatrix}$, and $\mathbf{T}_s = k_s \begin{bmatrix} 1 & l_{sa} & 0 & 0 \\ 1 & -l_{sa} & 0 & 0 \\ 0 & 0 & 1 & l_{sb} \\ 0 & 0 & 0 & -l_{sb} \end{bmatrix}$ with sensitivity of

displacement sensor $k_s$.

For the inertial principal axis, the dynamic functions of rotor shaft are expressed to

$$\begin{cases} m\ddot{x}_i = f_{xa} + f_{xb} \\ J_y\ddot{\beta}_i - J_z\Omega\dot{\alpha}_i = p_\beta = f_{xb}l_{ab} - f_{xa}l_{aa} \\ m\ddot{y}_i = f_{ya} + f_{yb} \\ J_x\ddot{\alpha}_i + J_z\Omega\dot{\beta}_i = p_\alpha = f_{ya}l_{aa} - f_{yb}l_{ab} \end{cases} \tag{15}$$

where $m$ is the mass, $J_y$ and $J_x$ are the equatorial moment of inertial, and $J_z$ is the polar moment of inertial.

The matrix form could be obtained as follows

$$\boldsymbol{M}\ddot{\boldsymbol{q}}_i + \boldsymbol{G}\dot{\boldsymbol{q}}_i = \boldsymbol{f} \tag{16}$$

where mass matrix is $\boldsymbol{M} = \mathrm{diag}[m, J_y, m, J_x]$, the coupling matrix is $\boldsymbol{G} = \begin{bmatrix} 0 & 0 & 0 & 0 \\ 0 & 0 & 0 & J_z\Omega \\ 0 & 0 & 0 & 0 \\ 0 & -J_z\Omega & 0 & 0 \end{bmatrix}$

, and the force matrix is $\boldsymbol{f} = \begin{bmatrix} 1 & 1 & 0 & 0 \\ l_{ab} & -l_{aa} & 0 & 0 \\ 0 & 0 & 1 & 1 \\ 0 & 0 & -l_{ab} & l_{aa} \end{bmatrix}\begin{bmatrix} f_{xb} \\ f_{xa} \\ f_{yb} \\ f_{ya} \end{bmatrix} = \boldsymbol{T}_f\boldsymbol{f}_m$.

By linearizing the magnetic forces expressed in (6) within the vicinity of equilibrium position [24], the magnetic forces of radial AMB are written into

$$\boldsymbol{f}_m = k_i\boldsymbol{i}_m + k_d\boldsymbol{q}_m \tag{17}$$

where the control current is $\boldsymbol{i}_m = [i_{ax}, i_{bx}, i_{ay}, i_{by}]$, $k_i$ is the current stiffness of radial AMB, and $k_d$ is the displacement stiffness of radial AMB.

### 3.2. Current Model of Rotor Shaft with Unbalance Terms

When the reference position of rotor shaft is 0 mm, the control current could be expressed as

$$\boldsymbol{i}_m = \textbf{Ctrl\_function}\left(0 - \boldsymbol{q}_s\right) \tag{18}$$

Substituting (14), (17) and (18) into (16), there is

$$\boldsymbol{M}\ddot{\boldsymbol{q}}_i + \boldsymbol{G}\dot{\boldsymbol{q}}_i = \boldsymbol{T}_f\left[k_i\textbf{Ctrl\_function}\left(-\boldsymbol{T}_s\boldsymbol{q}_g\right) + k_d\boldsymbol{T}_f^T\boldsymbol{q}_g\right] \tag{19}$$

with the control function $\textbf{Ctrl\_function}(s) = G_p(s)G_a(s)$.

The control function $G_p(s)$ of radial AMB is

$$\boldsymbol{G}_p\left(s\right) = \boldsymbol{T}_s\boldsymbol{G}_s\left(s\right)\boldsymbol{T}_s^{-1} \tag{20}$$

The control current $\boldsymbol{i}_r = [i_{rax}, i_{rbx}, i_{ray}, i_{rby}]^T$ of radial AMB is

$$\boldsymbol{i}_r\left(s\right) = \boldsymbol{G}_m\left(s\right)\left[-\boldsymbol{T}_s\boldsymbol{q}_g\left(s\right)\right] \tag{21}$$

The current control model of radial AMB is illustrated in Figure 9, and the whole model consisted of power amplification unit, H-bridge invertor, AMB model and current measurement unit. For the power amplification unit, $k_a$ is the proportional coefficient, and $k_v$ is the amplification coefficient. For the AMB model, $u_a$ is the voltage of EM winding, $L_a$ is winding inductance, $R_a$ is winding resistance, and $e_a$ is the electromotive force. For the current measurement unit, $k_{ie}$ is the gain of current sensor, and $k_{ic}$ is the feedback gain.

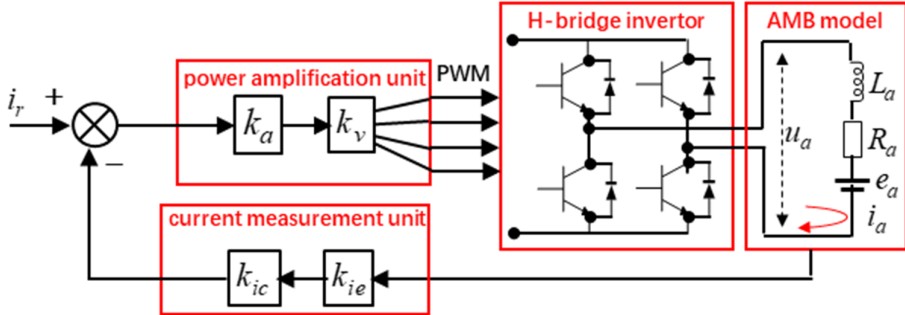

**Figure 9.** The power amplification model of radial AMB.

Based on the power amplification model in Figure 9, there are

$$\begin{cases} k_{amp}k_v\left(i_r - k_{ic}k_{ie}i_a\right) = u_a \\ L_a\dfrac{di_a}{dt} + R_a i_a + e_a = u_a \\ k_i\dfrac{dx_a}{dt} = e_a \end{cases} \tag{22}$$

The control current $i_m$ in (18) could be obtained as following

$$i_m(s) = k_a\frac{\omega_a}{s+\omega_a}i_r(s) - k_u k_a\frac{\omega_a s}{s+\omega_a}q_m(s) = G_{a1}(s)i_r(s) + G_{a2}(s)q_m(s) \tag{23}$$

with $k_a = \dfrac{k_{amp}k_v}{R + k_{amp}k_v k_{ic}k_{ie}}$ , $\omega_a = \dfrac{R + k_{amp}k_v k_{ic}k_{ie}}{L_a}$ , $k_u = \dfrac{k_i}{k_{amp}k_v}$ .

*3.3. Vibration Model of Rotor Shaft with Unbalance Terms*

As shown in Figure 8, in the generalized coordinate of rotor shaft, the unbalance terms of rotor shaft are

$$\Delta q = q_i - q_g = \begin{bmatrix} \varepsilon_s\cos\left(\Omega t + \psi_s\right) \\ \varepsilon_d\sin\left(\Omega t + \psi_d\right) \\ \varepsilon_s\sin\left(\Omega t + \psi_s\right) \\ -\varepsilon_d\cos\left(\Omega t + \psi_d\right) \end{bmatrix} \tag{24}$$

where $\varepsilon_s$ and $\psi_s$ are the amplitude and phase of static unbalance, respectively; and $\varepsilon_d$ and $\psi_d$ are the amplitude and phase of dynamic unbalance, separately.

The final dynamic function of rotor shaft controlled by the radial AMB is written into

$$\left(Ms^2 + Gs\right)\left[q_g(s) + \Delta q(s)\right] = -k_i\left[T_f T_s G_{a1}(s)G_s(s) + T_f T_f^T G_{a2}(s)\right]q_g(s) + k_d T_f T_f^T q_g(s) \tag{25}$$

Furthermore, for the dynamic function of rotor shaft with radial translation and rotating model, the dynamic model using the complex factor $j$ could be separated into

$$Trans(s) = f_x + f_y j = 2\left[k_d - k_i k_s G_{a1}(s)G_s(s) - k_i k_u s G_{a1}(s)\right]r_g \tag{26}$$

$$Rotate(s) = p_\alpha + p_\beta j = 2l_a\left[k_d l_a - k_i k_s l_b G_{a1}(s)G_s(s) - k_i k_u l_a s G_{a1}(s)\right]\theta_g \tag{27}$$

where $r_g = x_g + y_g j$, and $\theta_g = \alpha_g + \beta_g j$.

The transfer function $G_s(s)$ is simplified to solute the dynamic model of rotor shaft with unbalance term. Using the PID control model for radial AMB, (26) and (27) are expressed into

$$m\ddot{r}_g - m\varepsilon_s\Omega^2 e^{j(\Omega t + \psi_s)} = 2k_d r_g - 2k_i k_v k_s \frac{\omega_a}{s+\omega_a}\left[K_P r_g + \left(K_D + \frac{k_u}{k_s}\right)\dot{r}_g + K_I \int r_g dt\right] \quad (28)$$

$$J_x\ddot{\theta}_g - J_z\Omega\dot{\theta}_g j - \left(J_x - J_z\right)\varepsilon_d\Omega^2 e^{j(\Omega t + \psi_d)} =$$
$$2k_d l_a^2\dot{\theta}_g - 2l_a l_b k_i k_v k_s \frac{\omega_a}{s+\omega_a}\left[K_P\theta_g + \left(K_D + \frac{k_u l_a}{k_s l_b}\right)\dot{\theta}_g + K_I \int\theta_g dt\right] \quad (29)$$

where $K_P$, $K_I$ and $K_D$ are the proportional coefficient, the integration coefficient and the derivative coefficient.

The EM winding of the radial AMB would cut the magnetic lines when the rotor shaft works at high rotating speed, so the back electromotive force with a small amplitude would be generated [25], the control current would be increased. For the radial AMB at low speed, there are

$$\lim_{\Omega\to 0} r_a = 0$$
$$\lim_{\Omega\to\infty} r_a = \varepsilon_s \quad (30)$$
$$\lim_{\Omega\to\infty} \psi_{sg} = \psi_s + \pi$$

with

$$r_g = r_a e^{j(\Omega t + \psi_{sg})}$$
$$\begin{cases} r_a = \dfrac{m\varepsilon_s\Omega^2\sqrt{\omega_a^2 + \Omega^2}}{\sqrt{r_{a1}^2 + r_{a2}^2}} \\ \psi_{sg} = \psi_s + \arctan\dfrac{\Omega}{\omega_a} - \arctan\dfrac{r_{a2}}{r_{a1}} \end{cases} \quad (31)$$
$$r_{a1} = 2k_i k_v k_s K_P\omega_a - m\omega_a\Omega^2 - 2k_d\omega_a$$
$$r_{a2} = 2k_i k_v k_s K_D\omega_a\Omega - 2\frac{1}{\Omega}mk_i k_v k_s K_I\omega_a - m\Omega^3 - 2k_d\Omega$$

The rotating angle of rotor shaft is defined as following

$$\theta_g = \theta_a e^{j(\Omega t + \psi_{dg})}$$
$$\begin{cases} \theta_a = \dfrac{\sqrt{\left(J_x\omega_a\varepsilon_d\Omega^2 - J_z\omega_a\varepsilon_d\Omega^2\right)^2 + \left(J_x\varepsilon_d\Omega^3 - J_z\varepsilon_d\Omega^3\right)^2}}{\sqrt{\theta_{a1}^2 + \theta_{a2}^2}} \\ \psi_{dg} = \psi_d + \arctan\dfrac{\left(J_x - J_z\right)\Omega}{\left(J_x - J_z\right)\omega_a} - \arctan\dfrac{\theta_{a2}}{\theta_{a1}} \end{cases} \quad (32)$$
$$\theta_{a1} = \left(J_z - J_x\right)\omega_a\Omega^2 + 2\omega_a l_a\left(k_i k_v k_s l_b - k_d l_a\right)$$
$$\theta_{a2} = 2l_a l_b k_i k_v k_s\omega_a\left(K_D\Omega - \frac{K_P}{\Omega}\right) - 2\Omega k_d l_a^2 + \left(J_z - J_x\right)\Omega^3$$

Furthermore, there are

$$\lim_{\Omega\to 0} \theta_a = 0$$
$$\lim_{\Omega\to\infty} \theta_a = \varepsilon_d \quad (33)$$
$$\lim_{\Omega\to\infty} \psi_{dg} = \psi_d + \pi$$

The radial displacements measured by displacement sensors are expressed as

$$
\boldsymbol{q}_s = \begin{bmatrix} s_{ax} \\ s_{bx} \\ s_{ay} \\ s_{by} \end{bmatrix} = \boldsymbol{T}_s \begin{bmatrix} x_g \\ \beta_g \\ y_g \\ -\alpha_g \end{bmatrix} = \boldsymbol{T}_s \begin{bmatrix} r_a \cos\left(\Omega t + \psi_{sg}\right) \\ \theta_a \sin\left(\Omega t + \psi_{dg}\right) \\ r_a \sin\left(\Omega t + \psi_{sg}\right) \\ -\theta_a \cos\left(\Omega t + \psi_{dg}\right) \end{bmatrix}
$$

$$
= k_s \begin{bmatrix} \left(r_a \cos\psi_{sg} + \theta_a l_b \sin\psi_{dg}\right)\cos\Omega t + \left(-r_a \sin\psi_{sg} + \theta_a l_b \cos\psi_{dg}\right)\sin\Omega t \\ \left(r_a \cos\psi_{sg} - \theta_a l_b \sin\psi_{dg}\right)\cos\Omega t + \left(-r_a \sin\psi_{sg} - \theta_a l_b \cos\psi_{dg}\right)\sin\Omega t \\ \left(r_a \cos\psi_{sg} + \theta_a l_b \sin\psi_{dg}\right)\cos\left(\Omega t - \dfrac{\pi}{2}\right) + \left(-r_a \cos\psi_{sg} + \theta_a l_b \sin\psi_{dg}\right)\sin\left(\Omega t - \dfrac{\pi}{2}\right) \\ \left(r_a \cos\psi_{sg} - \theta_a l_b \sin\psi_{dg}\right)\cos\left(\Omega t - \dfrac{\pi}{2}\right) + \left(-r_a \cos\psi_{sg} - \theta_a l_b \sin\psi_{dg}\right)\sin\left(\Omega t - \dfrac{\pi}{2}\right) \end{bmatrix} \tag{34}
$$

The phase error between displacement terms $s_{ax}$ and $s_{ay}$ is $\pi/2$, and the amplitude of radial displacement terms $s_{ax}$ and $s_{ay}$ are expressed as follows

$$
\begin{aligned}
\left\| s_{ax} \right\| = \left\| s_{ay} \right\| = \sqrt{2r_a^2 + 2\theta_a^2 l_b^2 + 2r_a\theta_a l_b \sin\left(\psi_{dg} - \psi_{sg}\right)} \\
\left\| s_{bx} \right\| = \left\| s_{by} \right\| = \sqrt{2r_a^2 + 2\theta_a^2 l_b^2 - 2r_a\theta_a l_b \sin\left(\psi_{dg} - \psi_{sg}\right)}
\end{aligned} \tag{35}
$$

According to (35), the motion traces of rotor shaft at A-end and B-end are cycle trajectory, and

$$
\begin{cases}
\psi_{dg} - \psi_{sg} = n\pi \Rightarrow \left\| s_{ax} \right\| - \left\| s_{bx} \right\| = 0 \\
\psi_{dg} - \psi_{sg} = (n+0.5)\pi \Rightarrow \left\| s_{ax} \right\| - \left\| s_{bx} \right\| = \sqrt{2r_a^2 + 2\theta_a^2 l_b^2 + 2r_a\theta_a l_b} - \sqrt{2r_a^2 + 2\theta_a^2 l_b^2 - 2r_a\theta_a l_b}
\end{cases} \tag{36}
$$

Therefore, when the phase difference between the phase terms $\psi_{dg}$ and $\psi_{sg}$ is $\pi$, the amplitude error of displacement terms could be equal. When the phase difference between the phase terms $\psi_{dg}$ and $\psi_{sg}$ is $\pi/2$, the amplitude error of displacement terms reaches the maximum value.

## 4. Complex-Field Cross-Feedback Control of Magnetically Suspended Air-Blower

The stable suspension of the rotor shaft is a condition of active vibration control of the magnetically suspended air-blower. The cross-feedback control model was introduced to realize the decoupling control of the rotor shaft in translation and rotating DOFs. The transfer function of the cross-feedback control model is:

$$
G_{cr1}(s) = k_h \frac{s}{s + \omega_h} j\Omega - k_l \frac{\omega_l}{s + \omega_l} j\Omega \tag{37}
$$

where $k_h$ is the control coefficient for nutation, and $\omega_h$ is the cut-off frequency of high-pass filter, $k_l$ is the control coefficient for precession, and $\omega_l$ is the cut-off frequency of low-pass filter.

Furthermore, the cross-feedback control model with complex exponential function was introduced to realize the active vibration control of the magnetically suspended air-blower, and the transfer function is

$$
G_{cr2}(s) = k_h \frac{s}{s + \omega_h} e^{\theta_h j} \Omega - k_l \frac{\omega_l}{s + \omega_l} e^{\theta_l j} \Omega \tag{38}
$$

where $\theta_h$ is the phase of nutation, and $\theta_l$ is the phase of precession.

Since the sampling time of displacement signals could be defined by the control board, the phase difference $\pi/2$ between phase of nutation $\theta_h$ and precession $\theta_l$ could be set through the programming code. Compared to the normal cross-feedback control model, the complex exponential function in (38) has advantages of flexibly debugging and setting the lead phase.

Based on Equations (28) and (29), the unbalance force and vibration torque could generate reaction forces and torques to affect the stability and control precision of the rotor shaft. To suppress the unbalance vibration forces and torques, the rotor shaft should be forced to rotate around the inertial principal axis, through the Laplace transform, and the dynamic functions in (28) and (29) could be rewritten into

$$ms^2 r_i(s) = 2\left[k_d - k_i k_v k_s \frac{\omega_a}{s + \omega_a} G_c(s)\right]\left[r_i(s) - D(s)\right] \tag{39}$$

$$\left(J_x s^2 - J_z \Omega j s\right)\theta_i(s) = 2l_a\left\{k_d l_a - k_i k_v k_s l_s \frac{\omega_a}{s + \omega_a}\left[G_c(s) + G_{cr}(s)\right]\right\}\left[\theta_i(s) - I(s)\right] \tag{40}$$

where $r_i = x_i + y_i j$ and $\theta_i = \alpha_i + \beta_i j$, $G_c(s)$ is the control function.

The unbalance terms of the rotor shaft could be equivalent to the displacement deflection of the inertial principal axis, and the displacement terms $r_i$ and $\theta_i$ would be positively linear to the control force and torque, so the vibration force and torque of rotor shaft could be suppressed by eliminating the radial motions of the inertial principal axis. The functions in (39) and (40) could be expanded within the real number filed as following

$$f_x(s) = ms^2 x_i(s) = 2\left[k_d - k_i k_v k_s \frac{\omega_a}{s + \omega_a} G_c(s)\right]\left[x_i(s) - D(s)\right]$$

$$f_y(s) = ms^2 y_i(s) = 2\left[k_d - k_i k_v k_s \frac{\omega_a}{s + \omega_a} G_c(s)\right]\left[y_i(s) - D(s)\right] \tag{41}$$

and

$$J_x s^2 \alpha_i(s) + J_z \Omega s \beta_i(s) = 2l_a\left[k_d l_a - k_i k_v k_s l_s \frac{\omega_a}{s + \omega_a} G_{co}(s)\right]\left[\alpha_i(s) - I_\alpha(s)\right]$$

$$+ 2l_a k_i k_v k_s l_s \frac{\omega_a}{s + \omega_a} G_{si}(s)\left[\beta_i(s) - I_\beta(s)\right]$$

$$J_x s^2 \beta_i(s) - J_z \Omega s \alpha_i(s) = 2l_a\left[k_d l_a - k_i k_v k_s l_s \frac{\omega_a}{s + \omega_a} G_{co}(s)\right]\left[\beta_i(s) - I_\beta(s)\right] \tag{42}$$

$$- 2l_a k_i k_v k_s l_s \frac{\omega_a}{s + \omega_a} G_{si}(s)\left[\alpha_i(s) - I_\alpha(s)\right]$$

where

$$G_{co}(s) = G_c(s) + \frac{k_h \Omega s}{s + \omega_h}\cos\alpha_h - \frac{k_l \omega_l \Omega}{s + \omega_l}\sin\beta_l$$

$$G_{si}(s) = \frac{k_h \Omega s}{s + \omega_h}\cos\alpha_h - \frac{k_l \omega_l \Omega}{s + \omega_l}\sin\beta_l \tag{43}$$

and

$$\begin{cases} D_x(s) = \varepsilon_s \dfrac{s\cos\theta_s - \Omega\sin\theta_s}{s^2 + \Omega^2} \\ D_y(s) = \varepsilon_s \dfrac{s\sin\theta_s + \Omega\cos\theta_s}{s^2 + \Omega^2} \end{cases} ; \begin{cases} I_\alpha(s) = \varepsilon_d \dfrac{s\cos\theta_d - \Omega\sin\theta_d}{s^2 + \Omega^2} \\ I_\beta(s) = \varepsilon_d \dfrac{s\sin\theta_d + \Omega\cos\theta_d}{s^2 + \Omega^2} \end{cases} \tag{44}$$

As a conclusion, based on the dynamic function of the rotor shaft with the unbalance terms, there are:

(1)   The vibration force and torque caused by the unbalance terms of the rotor shaft having the same frequency as the rotating frequency.

(2)   The unbalance vibration forces in the radial direction are decoupling, but the

unbalance vibration torques are coupling in the radial direction.

(3) The coefficients $k_h$ and $k_l$ could suppress the nutation and precession vibration of the rotor shaft decoupling.

## 5. The Experimental Verification

### 5.1. Experimental System of Magnetically Suspended Air-Blower

The magnetically suspended air-blower is shown in Figure 10, and the whole system has the permanent magnet synchronous motor (PMSM), power supply system, main control unit (MCU), data acquisition system (DAQ) and axial flow fan. For the PMSM, the rotor shaft wats levitated by the thrust-disc AMB in the axial direction, the radial suspension of the rotor shaft was realized by magnetic bearings at the upper and lower end, and the protective bearings at upper and lower ends could restrict the large displacement deflection of the rotor shaft. The rotation of the rotor shaft around the axial principal axis was driven by the PMSM, and the rotating speed was regulated by a three-phase invertor. The power supply system could output the control voltage to axial and radial AMBs. The MCU based on a FPGA chip and a DSP chip could realize the program compiling of a negative-feedback control loop and complex-field cross-feedback control method, and then the control signals for the radial and axial AMBs were generated to tune the magnetic forces. The DAQ system based on the eddy current displacement sensors and the data acquisition board could collect the displacement signals of the rotor shaft, and other kinds of system signals, such as the control currents and rotating speed. were also measured by the DAQ system. In addition, the axial flow fan was used to lower the temperature of the magnetically suspended air-blower during the long-time high-speed operation. The system parameters of the magnetically suspended air-blower are listed in Table 1.

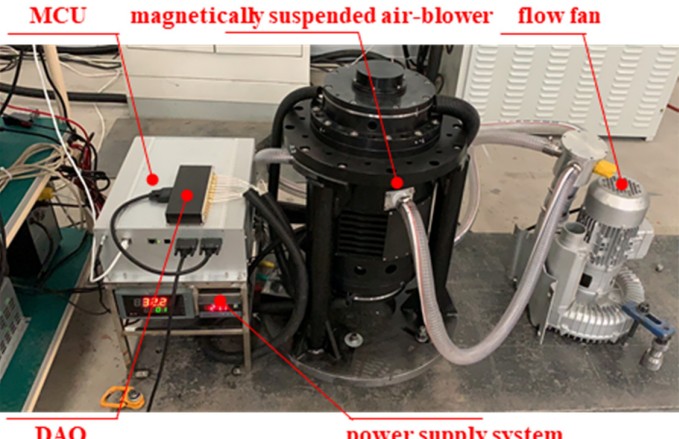

**Figure 10.** The magnetically suspended air-blower system.

**Table 1.** System Parameters of Magnetically Suspended Air-blower.

| Symbol | Quantity | Value |
|:---:|:---:|:---:|
| $m$ | mass of rotor shaft | 26 kg |
| $J_x$ | equatorial moment of inertia | 0.13 kgm² |
| $J_z$ | polar moment of inertia | 0.06 kgm² |
| $\Omega$ | rotational speed | 12,000 rpm |
| $k_{sx}$ | sensitivity of sensor | 9 V/mm |
| $k_{wx}$ | amplification coefficient | 0.2 A/V |

### 5.2. Rotor Trajectory of Magnetically Suspended Air-Blower

The suspension traces of the rotor shaft at the static state are plotted in Figure 11, and the initial positions [$x_a\ y_a\ x_b\ y_b$] of the rotor shaft in radial directions were 0.63 mm 0.68 mm −0.63 mm 0.65 mm. As illustrated in Figure 11a, the rotor shaft was suspended to the equilibrium positions in radial directions, and then the position values of the rotor shaft in radial directions were equal to zero. Moreover, the axis orbits of the rotor shaft at A-end and B-end were respectively plotted in Figure 11b,c, and the position terms of the rotor shaft at A-end and B-end were set at the steady-state value when it was suspended at center-position.

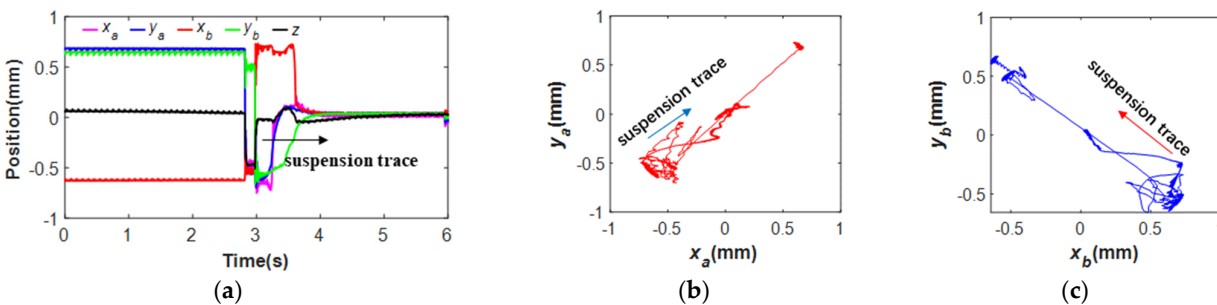

**Figure 11.** The suspension traces of rotor shaft at the static state, (**a**) the suspension displacements of rotor shaft in axial and radial directions, (**b**) the axis orbit of rotor shaft at A-end, (**c**) the axis orbit of rotor shaft at B-end.

The radial suspension traces of the rotor shaft during the rotating state are plotted in Figure 12. When the rotor shaft was located on the protective bearing, the displacement terms of the rotor shaft at A-end and B-end would be deflected from the steady-state value in Figure 12a, and then the rotor shaft was forced to suspend at the equilibrium position when the position term in the radial direction was zero. Moreover, the axis orbits of the rotor shaft at A-end and B-end during the rotating state are illustrated in Figure 12b,c. The suspension positions of the rotor shaft were forced back to the balanced positions.

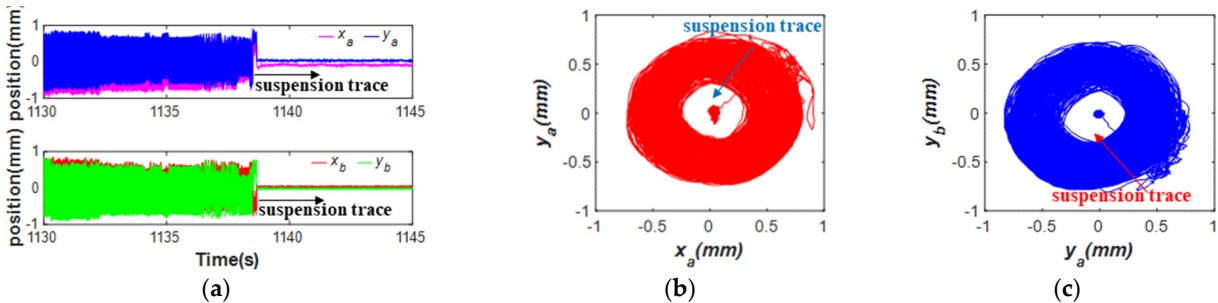

**Figure 12.** The suspension traces of rotor shaft at the rotating state, (**a**) the suspension displacements of rotor shaft in radial directions, (**b**) the axis orbit of rotor shaft at A-end, (**c**) the axis orbit of rotor shaft at B-end.

The dynamic positions of the rotor shaft at different rotating speeds are plotted in Figure 13. As shown in Figure 13a, when the rotating speed was 3000 rpm, the phase difference between the rotor displacement $x_a$ and $y_a$ at A-end of rotor shaft was $\pi/2$, and the phase difference at the B-end was also $\pi/2$. Moreover, the phase difference at A-end and B-end when the rotating speed was 6000 rpm is plotted in Figure 13b. For the theoretical expression in (34), the phase error between displacement terms $s_{ax}$ and $s_{ay}$ was $\pi/2$, and the phase difference shown in was still $\pi/2$, so the results of the theoretical analysis were verified.

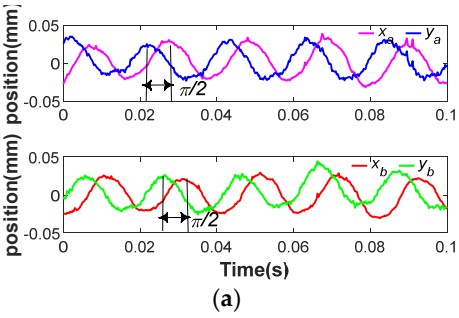
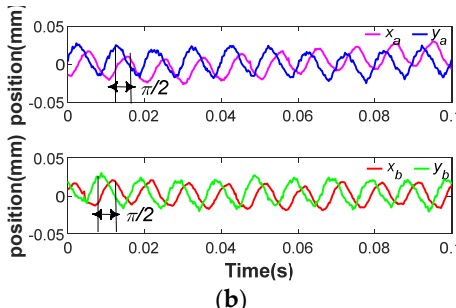

<div style="text-align:center">(<b>a</b>)      (<b>b</b>)</div>

**Figure 13.** (**a**) The phase difference of dynamic displacements at 3000 rpm, (**b**) the phase difference of dynamic displacements at 6000 rpm.

### 5.3. Vibration Characteristics of Magnetically Suspended Air-Blower

The vibration traces of the rotor shaft are illustrated in Figure14. The vibration amplitudes at A-end of the rotor shaft are plotted in Figure 14a, and those at B-end are shown in Figure 14b. The maximum vibration amplitude of rotor shaft was about 153 at the initial suspension point during the acceleration process, and the vibration amplitude of the rotor shaft was dampened with increase of rotating speed. When the rotating speed was approaching the rated value of 24,000 rpm, the vibration amplitude of rotor shaft reduced to 0.8 during the constant-speed process. For the deceleration process of the magnetically suspended air-blower, the vibration amplitude of the rotor shaft increased to 90 at the low rotating speed. Therefore, the vibration amplitude of the magnetically suspended air-blower with unbalance terms could be varied with rotating speed.

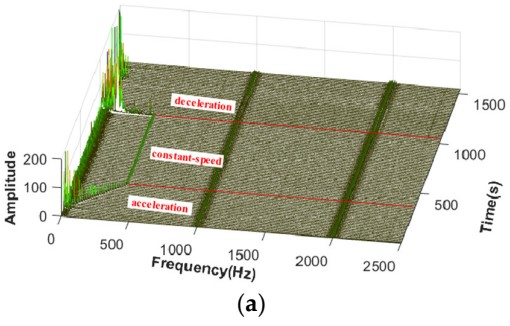
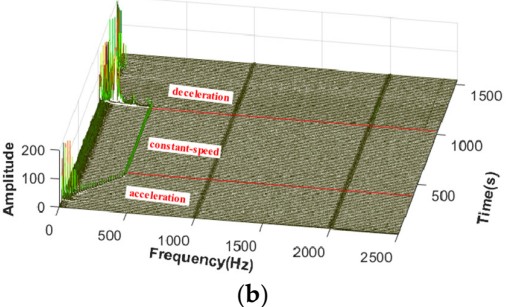

<div style="text-align:center">(<b>a</b>)      (<b>b</b>)</div>

**Figure 14.** The vibration amplitudes of rotor shaft during the speed regulation process, (**a**) the vibration amplitude of rotor shaft at A-end, (**b**) the vibration amplitude of rotor shaft at B-end.

### 5.4. Cross-Feedback Control of Magnetically Suspended Air-Blower

With the cross-feedback control of magnetically suspended air-blower, the nutation vibration and precession vibration of the rotor shaft could be effectively suppressed, and the root locus of nutation and precession are plotted in Figure 15. The root locus plotted by the red line indicates the high-frequency nutation vibration of rotor shaft, and the low-frequency precession vibration of rotor shaft is presented by the blue line. As shown in Figure 15a, the root locus of nutation vibration and precision vibration varied with the value of high-frequency nutation coefficient $k_h$. The imaginary value of nutation vibration was reduced with the value of $k_h$, and the real value increased with value of $k_h$, so the vibration amplitude of nutation was suppressed by increasing the value of $k_h$. Moreover, the root locus of precession vibration was also affected by the value of $k_h$, and vibration amplitude of precession vibration was mitigated by increasing the value of $k_h$. Furthermore, the root locus of nutation vibration and precession vibration choosing different values of $k_l$ are plotted in Figure 15b. The root locus of nutation vibration was not affected by the variation of $k_l$, and the vibration amplitude of precession was reduced

by increasing the value of $k_l$. Consequently, a greater value of $k_h$ could be used to suppress the nutation vibration of the rotor shaft, and this also mitigated the precession vibration.

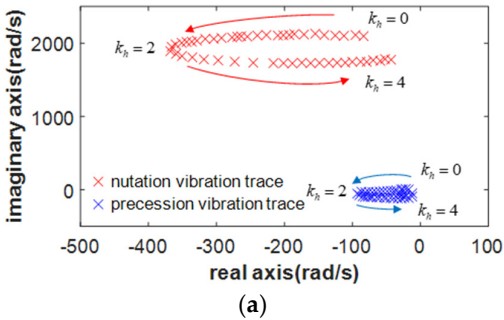
(a)

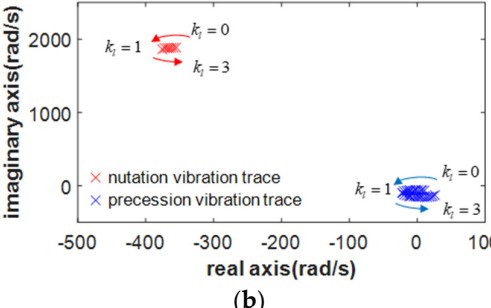
(b)

**Figure 15.** The root locus of nutation and precession vibration, (**a**) the relationship between root locus of nutation vibration and the value of $k_h$, (**b**) the relationship between root locus of precession vibration and the value of $k_l$.

Furthermore, the dynamic displacements and the power spectra of the rotor shaft in the magnetically suspended air-blower were measured, so the functions of the complex-field cross-feedback control model with different damping coefficients could be analyzed and verified. Firstly, the dynamic displacements of the rotor shaft at 12,000 rpm with different values of $k_h$ are shown in Figure 16a. The displacement curve of the rotor shaft without using the complex-field cross-feedback model is plotted as the green line, the displacement curve of the rotor shaft with the complex-field cross-feedback control model ($k_h = 0$ and $k_l = 1$) is shown by the red line, and the displacement curve of the rotor shaft with cross-feedback control model ($k_h = 4$ and $k_l = 1$) is marked by the blue line. The maximum value (peak-peak value) of dynamic displacement without applying the complex-field cross-feedback control model was 0.33 mm, and it reduced to 0.23 mm when the cross-feedback control model with $k_h = 0$ and $k_l = 1$ was applied to active control of magnetically suspended air-blower, and the maximum value of dynamic displacement was 0.11 mm when the coefficients were $k_h = 4$ and $k_l = 1$. Moreover, the power spectra of dynamic displacements with different coefficients are plotted in Figure 16b. In detail, the power spectrum density (PSD) without using the complex-field cross-feedback control model is shown by the green line, the PSD of precession vibration was 0.02, the PSD of synchronous vibration was 0.06, and the PSD of nutation vibration was 0.10. When the coefficients of complex-field cross-feedback control model were chosen as $k_h = 4$ and $k_l = 1$, the PSD of precession vibration reduced to 0.002, the PSD of synchronous vibration was 0.06, and the PSD of nutation vibration was 0.05. Therefore, based on the above analysis about the dynamic displacements and power spectrum of the rotor shaft with different parameters, the complex-field cross-feedback control was useful to suppress the nutation and precession vibrations, and the great nutation damping was more beneficial to control the nutation vibration.

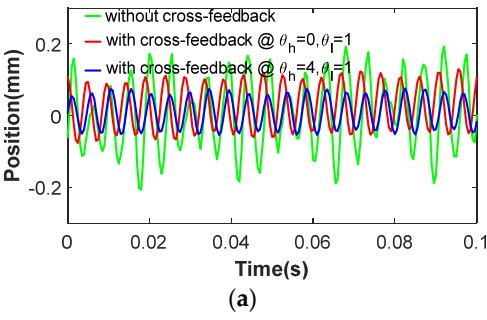 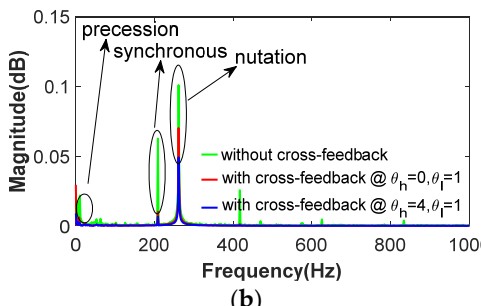

**Figure 16.** (**a**) The dynamic displacements of rotor shaft with different values of $k_h$ at 12,000 rpm, (**b**) the power spectrum of dynamic displacement with different values of $k_h$ at 12,000 rpm.

Similarly, the dynamic displacements and power spectrum of the rotor shaft with different values of $k_l$ are plotted in Figure 17, and the results indicated that the great value of $k_l$ could effectively suppress the precession vibration of the rotor shaft.

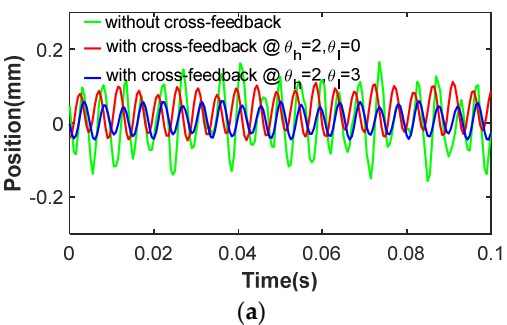 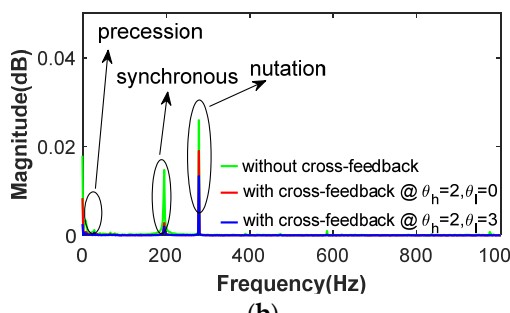

**Figure 17.** (**a**) The dynamic displacements of rotor shaft with different values of $k_l$ at 12,000 rpm, (**b**) the power spectrum of dynamic displacement with different values of $k_l$ at 12,000 rpm.

## 6. Conclusions

The vibration characteristics of the magnetically suspended air-blower were investigated in this article. Firstly, the structure and prototype of a magnetically suspended air-blower with radial and axial AMBs were introduced, and the displacement coordinates of the magnetically suspended air-blower were established. Moreover, the force models of AMB were developed using the equivalent magnetic circuit model, and then the dynamic models of the rotor shaft in the magnetically suspended air-blower were established. Furthermore, the vibration behaviors of the rotor shaft in the magnetically suspended air-blower were analyzed. Finally, the experimental results verified following conclusions:

(1) The displacement trajectory of the rotor shaft in *x* axis had the phase difference to the displacement in *y* axis.
(2) The vibration amplitude of rotor shaft in the magnetically suspended air-blower was affected by rotating speed.
(3) The complex-field cross-feedback control model was useful to suppress the vibration amplitudes of the magnetically suspended air-blower, and the high frequency damping coefficient could effectively mitigate the nutation and precession vibration.

Compared to other methods, the proposed vibration analysis more accurately describes the precession and nutation of a high rotating speed rotor shaft with magnetic forces. The designed control method is easy to implement for vibration control of the magnetically suspended air-blower.

**Author Contributions:** Data curation, L.Z.; Investigation, L.Z. and B.X.; Methodology, L.Z. and B.X.; Project administration, W.N.; Resources, W.N.; Supervision, W.N.; Validation, L.Z.; Writing—

original draft, L.Z.; Writing—review & editing, W.N. and B.X. All authors have read and agreed to the published version of the manuscript.

**Funding:** This research received no external funding.

**Data Availability Statement:** Not applicable.

**Conflicts of Interest:** The authors declare no conflict of interest.

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
