# Peer review of "Vibration Analysis and Active Control of Rotor Shaft in Magnetically Suspended Air-Blower"

_machines, doi:10.3390/machines10070570_

Round 1

Reviewer 1 Report

The current version is good for publication. The main contribution of the paper is that "the paper investigated  the vibration characteristics of rotor shaft in magnetically suspended air blower with unbalance terms" then they designed new control strategy to solve the problem. In my opinion, this is a very important contribution. The other parts are also good so I have no more questions on this paper.

Author Response

Thanks for your comment!

Reviewer 2 Report

The whole structure and logicality of this paper is quite good. The authors introduce the study background clearly and the figures in the paper well illustrate the study.

 However,

1-      The contributions should be highlighted in the Abstract,

2-       the authors should strengthen the significance analysis of the presented work in the Introduction,

3-       also should give the more comparative analysis with the existent results.

4-      Reduced the Self-citation for the last author.

Author Response

Thanks for your comment, and all revisions had been highlighted by the red texts in the manuscript.

  • The contributions should be highlighted in the Abstract,

Response: The contributions of manuscript are highlighted in abstract.

In the abstract on page 1:

The results indicate that the vibration analysis of rotor shaft is meaningful to the design and control of magnetically suspended air-blower.

  • the authors should strengthen the significance analysis of the presented work in the Introduction,

Response: The significance analysis of manuscript is added in the introduction.

In the introduction on page 2:

The vibration analysis is critical to structure design and active control high-speed rotor suspended by the magnetic forces, and it is fundamental to the vibration control of air-blower using the magnetic forces.

  • also should give the more comparative analysis with the existent results.

Response: The comparative analysis is added

In the conclusion on page 17:

Compared to other methods, the proposed vibration analysis is more accurately de-script the precession and nutation of high rotating speed rotor shaft with the magnetic forces. The designed control method is easy to be implemented for the vibration control of magnetically suspended air-blower.

4-Reduced the Self-citation for the last author.

Response: The self-citation of last author had been reduced.

Reviewer 3 Report

Hello Authors,

Please review the notes highlighted and created within the attached PDF document.

Thanks.

Author Response

Thanks for your comment! Please refers to the attached file, and all revisions are highlighted as the yellow texts .
